# Reflecting on Uncertainty Tolerance in Student Pharmacists Following an Inpatient Rounding Simulation: A Pilot Mixed-Methods Study

**DOI:** 10.3390/pharmacy13040111

**Published:** 2025-08-20

**Authors:** Nicole E. Cieri-Hutcherson, Collin M. Clark

**Affiliations:** Division of Education and Teaching Innovation, Department of Pharmacy Practice, School of Pharmacy and Pharmaceutical Sciences, University at Buffalo, New York, NY 14214, USA; collincl@buffalo.edu

**Keywords:** student pharmacist, pharmacy education, simulation, uncertainty, uncertainty tolerance

## Abstract

Background: With increasing involvement of pharmacists in clinical situations and expanding scope of practice, the expectation and management of uncertainty is a desirable component of pharmacy education, leading to improved uncertainty tolerance (UT) as pharmacists. Methods: The purpose of this pilot study was to determine whether an inpatient rounding simulation (IRS) that exposes student pharmacists to a level of uncertainty leads to changes in tolerance of ambiguity scale (TAS) scores and self-reflection comments. A pre-/post-observational, pilot survey study was conducted, including student pharmacists in their second professional year at the United States School of Pharmacy who were enrolled in an associated lab-based course in Spring 2024. Student teams completed a mock IRS and responded to clinical questions in a timed environment. Students completed pre-/post-simulation TAS and self-reflection on uncertainty/UT within the simulation. Pre-/post-simulation TAS scores were analyzed with a paired *t*-test. Qualitative analysis was used to identify themes in self-reflection. Results: 59 students responded (48% response rate). TAS was not different pre-/post-IRS (63.2 vs. 62.6, *p* = 0.63). When individual subscales were compared, a statistically significant difference was found in the insolubility subscale (10.5 vs. 9.5; *p* = 0.02). Themes of uncertainty that emerged during qualitative analysis of pre-simulation included the clinical question posed to the team. Student pharmacists overcame uncertainty pre-simulation by working with their team and relying on the availability of resources. Themes of uncertainty emerging during qualitative analysis of post-simulation included discerning the best answer. Student pharmacists overcame uncertainty post-simulation by working with their team and cited working with their team as a positive aspect of the IRS experience. Conclusions: In this pilot study, student pharmacists had high TAS scores both pre- and post-simulation. Students utilized their teams and cited this as a positive in an uncertain environment.

## 1. Introduction

Uncertainty tolerance (UT) has several definitions but is generally accepted to be the emotional, cognitive, and behavioral ways in which individuals respond to, minimize the negative effects of, and maximize the positive effects of uncertainty [1]. Demonstration of UT among physicians is viewed as a key competency of regulating bodies in several countries, including the United States [2]. With increasing involvement of pharmacists in clinical situations, arising from increasing prescriptive authority, collaborative practice agreements, and expansion of scope of practice, the expectation and management of uncertainty is a desirable component of pharmacy education. A focus on diagnosis in the provision of care has been highlighted in several areas of curricular elements in the United States (US) Accreditation Council for Pharmacy Education (ACPE) 2025 Standards [3,4]. There is evidence that physicians with low UT order more unnecessary diagnostic tests, are less likely to comply with evidence-based guidelines, and are more fearful in practice [5]. These traits can lead to increased healthcare costs and unnecessary tests and treatment. In addition, suppressing uncertainty may lead to a lack of consideration of alternative diagnoses, thereby increasing the likelihood of misdiagnosis. Therefore, addressing UT in healthcare providers, including healthcare providers in training, such as student pharmacists, who may be involved in diagnosis, may lead to reductions in diagnostic error, enhanced patient safety, and improved patient care.

In addition to pharmacists’ involvement in diagnostic approaches, UT in pharmacists would be desirable in other aspects of care. The use of medications off-label or differently than prescribed by patients may require pharmacists to maintain UT when managing patients in a variety of practice settings (e.g., community pharmacy, ambulatory care clinics, inpatient care). In addition, with the rapid approval of new medications, pharmacists must rapidly adapt to uncertainty in practice as care and guidelines shift to accommodate new approvals. Interprofessional team discussions may also introduce a degree of uncertainty.

Evaluation of interventions to improve UT in medical students has been conducted utilizing the Tolerance of Ambiguity Scale (TAS), Likert scales, and thematic analysis of focus groups [2]. Simulation-based activities, case-based activities, lecture with interactive workshops, and facilitated reflection training are among some of the interventions that have been studied in medical students [2,6,7,8,9,10,11,12]. In most cases, these interventions resulted in positive student experiences in time management, management of uncertainty, decision making, and improvements in TAS [2,6,7,8,9,10,11,12]. There is very limited evidence regarding UT in student pharmacists and its impact on patient care.

The objective of this pilot study was to assess whether an inpatient rounding simulation (IRS), which exposes student pharmacists to a level of uncertainty, led to changes in TAS scores, self-reflection comments, and thematic insights.

## 2. Materials and Methods

This study was conducted in accordance with the Declaration of Helsinki and approved by the Institutional Review Board of the University at Buffalo (STUDY00008163 on 31 March 2024). Second professional year (of four professional years) student pharmacists (n = 122) enrolled in a pharmaceutical care-based lab course offered in Spring 2024 within the required didactic curriculum of the University at Buffalo School of Pharmacy and Pharmaceutical Sciences (UB SPPS) were invited to participate in an optional research opportunity regarding UT via a course learning management system announcement and email as a convenience sample. The UB SPPS PharmD program consists of four professional years following pre-requisite coursework. Informed consent was obtained electronically prior to study participation. All students were offered extra course credit for completion of the study’s elements as an incentive. Students who did not complete the consent form could still obtain extra credit for completion of study elements, and responses were not included in this study. Completion of study materials did not impact the grade obtained in the IRS or associated course.

Pre-and post-IRS data for the TAS and self-reflection questions were collected via electronic survey software, SurveyMonkey. Pre-and post-IRS, student pharmacists completed the TAS. The Budner TAS was used to assess UT, which includes 16 items with a 7-point Likert scale in which students were asked to please respond to statements by indicating the extent to which they agree or disagree with them and choosing the number at the right that best represented evaluation of the item: SA = Strongly Agree; MA = Moderately Agree; A = Slightly Agree; N = Neither Agree Nor Disagree; D = Slightly Disagree; MD = Moderately Disagree; SD = Strongly Disagree [13]. Even-numbered questions are reverse-scored prior to the summation of scores. The TAS includes three sub-scales, including novelty, complexity, and insolubility, that can be used to identify the major source of intolerance of ambiguity. Self-reflection questions were generated by the authors. Pre-IRS students were also asked self-reflection questions regarding the following: 1. Reflect on what about the Inpatient Rounding Simulation do you believe will make you feel the most uncertain; 2. How do you plan to overcome feelings of uncertainty if experienced? 3. What plans do you have for when faced with these types of situations to manage feelings of uncertainty? and 4. When leaving a situation in which you felt uncertain, how did this make you feel? Reflect on both positive and negative feelings. The TAS was repeated post-IRS with the following self-reflection questions: 1. During the Inpatient Rounding Simulation, to what degree did you feel uncertain in recommendation making? (5-point Likert scale); 2. Reflect on what about the Inpatient Rounding Simulation and what made you feel the most uncertain; 3. How did you overcome feelings of uncertainty (if at all)? 4. What plans would you make for the future when faced with similar situations to manage feelings of uncertainty? 5. When leaving this situation in which you may have felt uncertain, how did this make you feel? Reflect on both positive and negative feelings.

### 2.1. Inpatient Rounding Simulation and Introduction of Uncertainty

Up to two weeks prior to the IRS, student pharmacists individually completed the pre-class TAS and self-reflection. There were two cohorts of students completing the IRS across two weeks. Students were also provided three clinical cases and review cases in teams; the clinical questions that were to be posed to student teams in class were unknown at this time. Students were asked to review a short video and a class plan to orient them to the IRS and grading elements.

In the IRS, student teams were presented with simulated inpatient rounding on the three clinical cases previously provided. One clinical question regarding the case was posed to the student team by a mock interprofessional team played by teaching assistants. Student teams spent 5 min in the mock patient room. Student teams then proceeded to an adjacent team room to answer the question using any resources, electronic or otherwise, for 10 min. The clinical question and team room time were repeated for each of the three clinical cases (45 min total). Student teams turned in one answer sheet with the team’s answers to three clinical case questions. Student teams are graded based on the accuracy of their answers and the appropriateness of references used.

Student pharmacists individually completed the post-class TAS and self-reflection within one to two weeks of the IRS.

### 2.2. Quantitative Analysis

Individual items and total TAS scores were assessed descriptively and presented as medians and interquartile ranges (IQR). Pre-and post-simulation TAS scores were analyzed using a Wilcoxon signed-rank test. The three sub-scales of the TAS, including novelty, complexity, and insolubility, were also assessed descriptively and analyzed with a Wilcoxon signed-rank test. A *p*-value of <0.05 was considered statistically significant. An effect size (*r*) was calculated by dividing the z-statistic by the square root of the number of paired comparisons. Cohen’s criteria were applied in the interpretation of effect size, with less than 0.2 considered very small and meaningless; effect sizes greater than 0.2 and up to 0.5 were considered small but meaningful; effect sizes greater than 0.5 and up to 0.8 were considered medium; and effect sizes greater than 0.8 were considered large [14].

### 2.3. Qualitative Analysis

Qualitative analysis was employed using a rapid analytic approach to identify themes in self-reflection question responses [15]. The first analytic step was the summarization of student responses via a template matrix. Next, the data was organized into broad themes, and supporting quotations were identified. Each researcher independently coded the data to identify themes. The researchers met to review the themes and establish the rigor and validity of the coding through further discussion and adjustment.

## 3. Results

Of the student pharmacists enrolled in the associated course, 59 responded and completed informed consent (48% response rate).

### 3.1. Quantitative Data

Median (interquartile range) results per item on the TAS have been reported in Table 1.

Comparison of TAS scores is presented in Table 2. The overall TAS score did not differ pre- and post-IRS. Of the three subscales, the novelty subscale had the highest relative score. When individual subscales were compared, a statistically significant difference was found only in the insolubility scale (10.5 vs. 9.5; *p* = 0.02).

### 3.2. Qualitative Data

Pre-IRS, the following themes of aspects of uncertainty were identified by students on their individual responses: questions/additional information given about the cases, case complexity, not knowing everything or having a lack of experince and knowledge, time limits, references and finding resources, feeling “unprepared”, discerning the best answer, navigating group disagreement, lack of information, the acuity of the event, fear of making a mistake, adverse effects and applying knowledge to practice. Post-IRS, student responses revealed the same themes of aspects of uncertainty.

Thematic analysis revealed that participants frequently emphasized the importance of working with their team following the IRS to overcome uncertainty (Table 3). Pre-IRS, student pharmacists identified the following ways of overcoming uncertainty: studying or preparing, managing preexisting mental illness, exposure to topic/situation, unknown, working with team; having resources and experience with a literature search, adaptibility, taking their time, asking questions, focusing on items within control/admitting when they do not know the answer, staying calm, being self-confident, keep on trying, improvising. Post-IRS, student responses revealed the same themes of ways to overcome uncertainty.

## 4. Discussion

There is a paucity of literature evaluating interventions for managing and improving UT in student pharmacists and pharmacists. As such, intervention and evaluation approaches regarding UT for student pharmacists will be crucial in informing student training. Currently, there are no established best practices for recognizing uncertainty or improving UT. Parts of managing uncertainty include the ability to recognize its presence in factual, emotional, or skills-based situations, communicating its presence, and using uncertainty to trigger next action planning. With an increased focus from ACPE’s Standards 2025 on patient assessment and diagnosis, UT may be an essential component of practice in the future.

The TAS contains a sliding scale ranging from 16 to 112, with lower scores indicating higher ambiguity (uncertainty) tolerance [13]. While there is no established cut score or strict interpretation of TAS scores, prior studies indicate healthcare providers with a TAS score between 44 and 52 are tolerant of uncertainty [16]. While a significant difference in TAS score was not observed when comparing pre- and post-IRS student responses, student pharmacists’ scores tended to be higher than other healthcare providers’ scores, indicating a low level of UT. This result is not surprising given the precise nature of pharmacy training. The relatively higher scores on the novelty subscale suggest that students were uncomfortable in an unfamiliar situation. Given a historical role as an educator, rather than a decision-maker for patients, pharmacists may have difficulty adapting to the evolving role of the pharmacist in healthcare [17,18]. Studies investigating pharmacist decision-making have found that pharmacists tend to defer to others’ authority to avoid decision-making (problem-solving) and potential conflicts, a characteristic that limits progress toward an expanded scope of practice. Insolubility is the quality of being very difficult or impossible to solve.

Although there was no significant difference in the TAS pre- and post-IRS, the insolubility sub-scale of the TAS demonstrated a statistically significant improvement from pre- to post-IRS, suggesting that student pharmacists’ ratings of tolerance for problems being very difficult improved following the IRS. This improvement suggests that the IRS, including complex cases that may have been perceived as more challenging than typical cases, assisted students in their ability to manage insolubility. Student scores on the novelty and complexity sub-scales did not change significantly. Larger studies with repeated exposure may lead to improvements in other sub-scales of the TAS.

A range of intervention types has been studied for the assessment of uncertainty, particularly in the literature pertaining to the training of medical students. A scoping review of interventions to improve UT in medical students included three studies utilizing simulation-based training to incorporate elements of uncertainty, such as the use of a simulated patient or a clinical scenario [2]. Cognitive and behavioral responses of medical students were positive in all three simulation studies. Two of these studies reported negative emotional responses to the uncertainty introduced in the simulation. Student pharmacists’ response to uncertainty may be impacted by the type of stimulation or intervention implemented. In the scoping review, several studies reported a negative emotional response alongside a positive cognitive or behavioral response to an uncertainty element [2]. This was thought to be due to the high-stakes nature of an intervention, such as a highly weighted or graded simulation or assessment. Alternatively, a negative emotional response from students may be observed when only a single intervention is implemented. Careful consideration should be given to the nature of the assessment that is paired with the uncertainty element. Low-stakes or ungraded assessments paired with an uncertainty element and repeated exposure, such as a teaching or practice assessment followed by a repeat assessment, may enhance the student experience and improve psychological safety, allowing for growth in UT. As this pilot study evaluated a single assessment associated with uncertainty, student pharmacists may have had a negative emotional experience. In addition to repeating this assessment throughout the didactic curriculum to avoid negative emotional experiences, the experiential curriculum may also be utilized in the future. Experiential learning theory is a cornerstone for pharmacy education [3]. Experiential learning inherently incorporates uncertainty and may be an additional opportunity to explore UT in student pharmacists.

Thematic responses pre- and post-IRS indicated that teamwork played a role in the ability to manage uncertainty. Team-based learning improves communication, critical thinking, and effective preparation to manage clinical situations with increased confidence [19]. This will be an interesting element to evaluate further in future studies. Qualitative data supported that the clinical question posed to the team was expected to be an uncertain element of the IRS, which reinforces this design element. Other sources of uncertainty included the case itself, time, preparation, acuity, ability to apply knowledge, and fear of making mistakes. Although the self-reflection questions did not specifically address whether teamwork could have reduced these sources of uncertainty, many could be addressed through teamwork, including disseminating the burden of preparation, use of group knowledge, reliance on peers to improve efficiency in time constraints, and reducing fear through support systems. For future studies, additional reflection on the role of the team in managing uncertainty will be added, including how it added to the psychological safety of the students in the IRS.

One of the required ACPE curricular educational outcomes for the pharmacy graduates is “Problem solving process (Problem-Solver).” [3] The ACPE defines Problem-Solver as “The graduate is able to use problem solving and critical thinking skills, along with an innovative mindset, to address challenges and to promote positive change.” How pharmacy curricula address this educational outcome varies [20]. An IRS introducing uncertainty elements may be a viable way to measure the problem-solving process. Although student pharmacists completed the pre- and post-IRS TAS and self-reflection individually, the team approach to decision-making and reflective practice creates a psychological safety for the student, allowing them to grow in their UT. The IRS focused on self-reflection of UT may also address additional educational outcomes, including attitudinal elements of self-awareness and professionalism [3].

This pilot study is not without limitations, and caution should be taken when applying the results found. Self-reflection responses were derived from a small sample of students from one US institution, limiting external validity. UT may have been influenced by the faculty, curriculum, or other factors linked to the specific school of pharmacy, which threaten the internal validity of this study. While students served as their own matched controls, the lack of a comparator group for other types of interventions incorporating uncertainty limits this study, as well. Several tools are available to assess uncertainty; however, the TAS has not been validated in this population. McLain’s MSTAT-II or Geller’s Tolerance for Ambiguity Scale may have been useful as either the alternative or supplementary methods of assessing uncertainty [21,22]. This study asked students about their perceptions of uncertainty at the time of the simulation, and post-IRS may have been completed up to two weeks following the IRS. As such, responses may not be indicative of the student experience within or immediately following the IRS and are subject to recall bias. Future iterations of the pilot study will have students complete the assessments immediately following the IRS. The response rate of 48% increases the chance of non-response bias. Potential biases may have occurred throughout the coding process. While we used multiple coders, no formal assessment of interrater reliability was calculated. For future studies, the incorporation of simulation, highlighting uncertainty, should continue throughout multiple time points in the didactic and experiential curriculum. Evaluation of student responses as they progress through the didactic and experiential curriculum can then be compared for changes in TAS and uncertainty, while limiting negative experience. Although this study did not address the relationship between UT and clinical accuracy of evidence-based decision-making and diagnostics, it would be an interesting area to evaluate in the future, comparing TAS with the likelihood of correct clinical recommendations in IRS.

As a pilot, valuable insights were gained for future directions in assessing UT in the curriculum. The pharmacy curriculum at this US institution has recently been updated, and the inclusion of UT assessment within the curriculum will continue. The research team will be assessing students across multiple years of training to compare progression in UT and Advanced Pharmacy Practice Experience readiness. Addressing how UT impacts stress and wellbeing, as well as the impact of demographic factors such as gender and year of training, will be interesting items to include as the project is scaled up [23].

## 5. Conclusions

In this pilot study, student pharmacists demonstrated high TAS scores both before and after the IRS. This suggests that UT was low both before and after the IRS. Qualitative findings highlight the dependence of student pharmacists on their teams, and student pharmacists cited this as a positive in an uncertain environment. Future studies with repeated UT assessment throughout the curriculum, relating self-reflection to teamwork, and linking UT with the accuracy of clinical recommendations are needed.

## Figures and Tables

**Table 1 pharmacy-13-00111-t001:** Tolerance of Ambiguity Scale Items Pre-and Post-IRS.

TAS Item	Pre-IRS, Median (IQR)	Post-IRS, Median (IQR)
An expert who does no’t come up with a definite answer probably does no’t know too much.	3 (2, 4)	2 (2, 3)
I would like to live in a foreign country for a while ^a^.	3 (2, 6)	3 (2, 5)
There is really no such thing as a problem that canno’t be solved.	5 (3, 6)	4 (3, 6)
People who fit their lives to a schedule probably miss most of the joy of living ^a^.	5 (3, 6)	5 (3, 6)
A good job is one where what is to be done and how it is to be done are always clear.	4 (3, 5)	5 (3, 6)
It is more fun to tackle a complicated problem than to solve a simple one ^a^.	3 (2, 4)	3 (2, 4)
In the long run, it is possible to get more done by tackling small, simple problems rather than large and complicated ones.	5 (4, 6)	5 (4, 6)
Often, the most interesting and stimulating people are those who do no’t mind being different and original ^a^.	2 (1, 3)	2 (1, 3)
What we are used to is always preferable to what is unfamiliar.	5 (4, 6)	5 (4, 6)
People who insist upon a yes or no answer just do no’t know how complicated things really are ^a^.	3 (2, 4)	3 (2, 4)
A person who leads an even, regular life in which few surprises or unexpected happenings arise really has a lot to be grateful for.	4.5 (4, 5)	4 (4, 5)
Many of our most important decisions are based upon insufficient information ^a^.	3 (2, 4)	3 (2, 3)
I like parties where I know most of the people more than ones where all or most of the people are complete strangers.	6 (6, 7)	6 (5, 7)
Teachers or supervisors who hand out vague assignments give one a chance to show initiative and originality ^a^.	3 (3, 5)	4 (3, 5)
The sooner we all acquire similar values and ideals, the better.	3 (2, 4)	3 (2, 5)
A good teacher is one who makes you wonder about your way of looking at things ^a^.	5 (3, 6)	5 (3, 6)

^a^ Indicates an item that is reverse scored (1 = 7, 2 = 6, 3 = 5, 4 = 4, 5 = 3, 6 = 2, 7 = 1).

**Table 2 pharmacy-13-00111-t002:** Tolerance of Ambiguity Scale Scores Pre- and Post-IRS.

TAS Item	Pre-IRS, Median (IQR)	Post-IRS, Median (IQR)	*p*-Value	r ^d^
TAS	63.2 (60, 68)	62.6 (58, 66)	0.63	0.11
Sub-scales				
Novelty ^a^	18.9 (17, 21)	18.4 (16, 21)	0.52	0.01
Complexity ^b^	33.9 (30, 37)	34.7 (32, 37)	0.19	0.20
Insolubility ^c^	10.5 (8, 13)	9.5 (8, 11)	0.02	0.01

^a^ Includes items 2, 9, 11, and 13. ^b^ Includes items 4, 5, 6, 7, 8, 10, 14, 15, and 16. ^c^ Includes items 1, 3, and 12. ^d^ r = z-statistic divided by the square root of N.

**Table 3 pharmacy-13-00111-t003:** Qualitative Thematic Analysis.

Self-Reflection Item	Rapid Analysis Response Theme	Illustrative Quotation
Aspect of uncertainty pre-IRS	Clinical question posed to the student team	“I feel the most uncertain about what questions my group will be asked, and whether our answers will be what the instructor is looking for.” “Any uncertain variable in the simulation will make me feel unsure, but most likely if I get asked a question I don’t know the answer to.”
Way of overcoming uncertainty pre-IRS	Working with their team and Availability of resources	“Speaking with my team and weighing out the risks and benefits.” “I would ask my peers on their thoughts and come up with an agreed solution.” “Have resources prepared ahead of time.” “I overcome feelings of uncertainty by familiarizing myself with potential resources I can use to answer the questions. “I would look over the case to see what diseases/drugs we are treating and search guidelines that could help me answer the potential questions.”
Aspect of uncertainty post-IRS	Discerning the best answer	“I think what made me feel the most uncertain was there being multiple viable answers.” “I was most uncertain about recommending the right thing. I feel as if there are multiple studies out there to back up different things.”
Way of overcoming uncertainty post-IRS	Working with their team	“Utilizing my group members’ input helped to overcome feelings of uncertainty.” “We overcame them by discussing possibilities of answers as a group and coming to an agreement on an answer.”
Positive aspects of the experience post-IRS	Working with their team	“I liked being able to work with the team to flush things out. I did not like not being able to clarify the information we were given.” “Collaborating with the team made me feel cooperative, and the discussions helped us make better decisions for the patients, but we were unsure if these decisions were right for the patients.”

## Data Availability

All data are contained within this paper.

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
