# Peer review of "Reflecting on Uncertainty Tolerance in Student Pharmacists Following an Inpatient Rounding Simulation: A Pilot Mixed-Methods Study"

_pharmacy, 2025, doi:10.3390/pharmacy13040111_

Round 1
Reviewer 1 Report
Comments and Suggestions for Authors
Thank you for inviting me to review this paper which explores pharmacy students’ tolerance of uncertainty. Unfortunately, I do not think the paper is currently suitable for publication and whilst this it is just a pilot study, there are key concerns in the methods which significantly undermine its credibility. The TAS has been used and there are some studies which have used it, including in health and the authors cite one of these; there is also some belated recognition of TAS limitations but currently the methods and results have several key concerns (see below for details). I am not sure if these can be addressed as several seem to relate back to the data collection (eg the qualitative aspect is incredibly basic, and I suspect linked to getting pharmacy students to write down responses to 4 pre- and 5 post- activity self-reflection questions. As would happen on a survey, open response qualitative questions are typically very poorly populated and this in my experience makes analysis a very difficult task. There are key issues in the TAS analysis and presentation (was reverse scoring used etc?) and some basic things about reporting the data also which gave concern.
Below specific comments are provided but I think overall, this paper would require very considerable improvements even in reporting pilot data; this is a potentially interesting area and the authors do have a promising area to consider but currently this pilot study contains too many concerns to be of publication standard. I hope these and comments below are of help.
Abstract – clear but could have given more detail of the pharmacy student sample by country at least. Did not think numbers were needed before the sections.
Introduction
This section was brief but ok and did just enough to orientate but did lack key detail like what interventions and things helped medical students.
P1l39 should this read ‘UT among physicians’.
P2 l44 – add US before ACPE to clarify where these standards apply for an international audience
P2l45 ‘There is evidence that physicians with […] are less likely’
P2l46-49 needs supporting references/evidence as multiple claims are made here. And also as the last sentence in this paragraph (which is very long) is based on this prior claim and will again feel unsupported if there is no link to specific evidence/literature.
P2l51-52 ‘involved in diagnosis’ was grammatically confusing – did this link back to pharmacy students and so should it need ‘who may be’ added?
P2l54-56 – have these UT interventions worked in medical students?
P2l57 – again for international audience explain a bit more about IRS and what this involved in introduction.
Methods
This section reported on the use of a validated Budner TAS but gives an unusual level of detail in the wrong places. It would be helpful to reflect on some of the psychometric properties of the TAS as it has been criticised for its α = .49 internal consistency etc. There are also others and some justification of this 1962 TAS would be welcome. Also, clarifying there is no formal cut-off ie what score would suggest someone is tolerant of ambiguity, or as noted later what the 3 sub-sales relate to. There was also no justification for the use of the self-reflection questions pre- and post IRS. (where did these 4+5 questions come from? Literature? The authors?) How did the students complete this also? Paper copies and written responses as opposed to verbally which these self-reflection questions might be better suited for? Also, the IRS is a group exercise and not clear what each student’s role is – ie are they meant to be working as a group, and if so, might this impact on their TAS. When I am teaching students, my sense is that group activities spread the load/risk/responsibility and so asking an individual student about a group task and assuming this is capturing their (possible) change in UT is not an ideal study design.
The qualitative analysis has some detail but lacks any supporting reference to recognised qualitative analysis literature. This aspect was not at the level required for credible qualitative reporting although it is recognised that asking students to write open responses to the 4+5 self-reflection questions might not give very rich responses.
P2l63-68 – 51 word length sentence is not appropriate. Also in this sentence what is second professional year and pharmaceutical care-based lab course? Again for non-US readers this might be helpful to know exactly at what stage these pharmacy students are at, as there is a large literature on the changes and socialisation and professionalisation that takes place in health care education. Perhaps even just something more to clarify if these are in their xth year of a total of y years on a standard US pharmacy degree.
P2l69-74 had too much detail about very specific things like the exact wording of the 7 Likert responses – this is not needed in a methods.
P38l89 and p3103 – how long was given for the pre- TAS and self-reflection? Post- was up to 2 weeks but not mentioned for pre-. On the post- test, why was up to 2 weeks allowed? Could this then be influenced by recall bias?
Results
This was not an appropriate results section. There is only one sentence referring the reader to a large table; the table reports median values but does not appear to alert the reader to the fact that even-numbered questions should be reverse scored ie they are worded to elicit the opposite response to odd-numbered questions, so this table may well confuse. Also in the table, an IQR is reported but is not correct, and it is Q3 and Q1 medians which are being report, and IQR (ie range) is Q3-Q1 so (2,4) should be reported as 2 as the IQR.
Table 2 does not help the reader either in explaining the sub-scales and these are simply presented without any context. Again Q3 and Q1 quartile integer values are given not an actual IQ range. A p value is also presented for these but with no methods detail in how this was calculated or what the cut-off for statistical significance (eg p=0.05) was.
The qualitative aspect is also not suitable for publication; as suspected, the student’s presumably written responses were probably very brief and resulted in just basic words or phrases being listed; table 3 gives a summary of these and quantifies them (the latter not being appropriate for qualitative analysis). There are occasional uses of quotation marks but nothing that would resemble a quotation as understood in the qualitative literature.
There was also not demographic detail or related analysis of whether any other independent variables might impact on tolerance of ambiguity.
Discussion
This is a very long section relative to the others and in fact it did contain much more helpful context and reflections on the TAS and also some other insights into pharmacy and wider healthcare education. However, based on the concerns raised above, caution is needed in trying to discuss the findings given the concerns identified in the methods and results.
Author Response
Reviewer 1:
- Abstract – clear but could have given more detail of the pharmacy student sample by country at least. Did not think numbers were needed before the sections.
- These changes have been made per reviewer recommendation
- This section was brief but ok and did just enough to orientate but did lack key detail like what interventions and things helped medical students.
- This section was bolstered to describe some of the successful interventions in medical students.
- P1l39 should this read ‘UT among physicians’.
- Added per reviewer recommendation
- P2 l44 – add US before ACPE to clarify where these standards apply for an international audience
- Added per reviewer recommendation
- P2l45 ‘There is evidence that physicians with […] are less likely’
- Added per reviewer recommendation
- P2l46-49 needs supporting references/evidence as multiple claims are made here. And also as the last sentence in this paragraph (which is very long) is based on this prior claim and will again feel unsupported if there is no link to specific evidence/literature.
- The claims in this section are is all from reference 5
- P2l51-52 ‘involved in diagnosis’ was grammatically confusing – did this link back to pharmacy students and so should it need ‘who may be’ added?
- Added per reviewer recommendation
- P2l54-56 – have these UT interventions worked in medical students?
- As above, this section was bolstered to describe interventions and success.
- P2l57 – again for international audience explain a bit more about IRS and what this involved in introduction.
- The IRS is described in extensive detail in the methods section. This is specific to the institution and not something conducted at every US school of pharmacy and as such the authors have chosen to keep the description in the methods.
- This section reported on the use of a validated Budner TAS but gives an unusual level of detail in the wrong places. It would be helpful to reflect on some of the psychometric properties of the TAS as it has been criticised for its α = .49 internal consistency etc. There are also others and some justification of this 1962 TAS would be welcome. Also, clarifying there is no formal cut-off ie what score would suggest someone is tolerant of ambiguity, or as noted later what the 3 sub-sales relate to. There was also no justification for the use of the self-reflection questions pre- and post IRS. (where did these 4+5 questions come from? Literature? The authors?) How did the students complete this also? Paper copies and written responses as opposed to verbally which these self-reflection questions might be better suited for? Also, the IRS is a group exercise and not clear what each student’s role is – ie are they meant to be working as a group, and if so, might this impact on their TAS. When I am teaching students, my sense is that group activities spread the load/risk/responsibility and so asking an individual student about a group task and assuming this is capturing their (possible) change in UT is not an ideal study design.
- Detail was added to this section to address reviewer concerns, the incentives, the way the survey was distributed, and the generation of the survey questions. Students worked in teams in the IRS, this is described in section 2.1.
- The qualitative analysis has some detail but lacks any supporting reference to recognised qualitative analysis literature. This aspect was not at the level required for credible qualitative reporting although it is recognised that asking students to write open responses to the 4+5 self-reflection questions might not give very rich responses.
- Detail added and edited in this section. Reference added for the method used.
- P2l63-68 – 51 word length sentence is not appropriate. Also in this sentence what is second professional year and pharmaceutical care-based lab course? Again for non-US readers this might be helpful to know exactly at what stage these pharmacy students are at, as there is a large literature on the changes and socialisation and professionalisation that takes place in health care education. Perhaps even just something more to clarify if these are in their xth year of a total of y years on a standard US pharmacy degree.
- Information added to the methods to describe the year of training of the student pharmacists.
- P2l69-74 had too much detail about very specific things like the exact wording of the 7 Likert responses – this is not needed in a methods.
- The authors chose to retain this section of the methods.
- P38l89 and p3103 – how long was given for the pre- TAS and self-reflection? Post was up to 2 weeks, but not mentioned for pre. On the post- test, why was up to 2 weeks allowed? Could this then be influenced by recall bias?
- Information was added about the timing of the pre- and post-surveys. Recall bias is discussed in the limitations but not called out specially with the term “recall bias” so this was added.
- Up to two weeks were allowed as students were split into two groups that completed the IRS during two separate weeks. Some students would have had only 1 week to complete the post-IRS, while others would have had two.
- This was not an appropriate results section. There is only one sentence referring the reader to a large table; the table reports median values but does not appear to alert the reader to the fact that even-numbered questions should be reverse scored ie they are worded to elicit the opposite response to odd-numbered questions, so this table may well confuse. Also in the table, an IQR is reported but is not correct, and it is Q3 and Q1 medians which are being report, and IQR (ie range) is Q3-Q1 so (2,4) should be reported as 2 as the IQR.
- We opted not to repeat results in the text of the results section we felt was better displayed in a table format.
- We have added to the Methods section that even numbered items are reverse coded as well as a footnote within Table 1.
- Providing the quartiles allows the reader to get a better understanding of the data spread vs. presenting a single number.
- Table 2 does not help the reader either in explaining the sub-scales and these are simply presented without any context. Again Q3 and Q1 quartile integer values are given not an actual IQ range. A p value is also presented for these but with no methods detail in how this was calculated or what the cut-off for statistical significance (eg p=0.05) was.
- We have added an explanation of the sub-scales in the Methods section.
- As above, we feel there is greater utility in providing the quartile values to the reader to interpret.
- We have added our cutoff for statistical significance to the methods.
- The qualitative aspect is also not suitable for publication; as suspected, the student’s presumably written responses were probably very brief and resulted in just basic words or phrases being listed; table 3 gives a summary of these and quantifies them (the latter not being appropriate for qualitative analysis). There are occasional uses of quotation marks but nothing that would resemble a quotation as understood in the qualitative literature.
- Student responses were not basic words or very brief. Quoted aspects were added to the revised Table 2
- We have updated and improved our qualitative approach and reporting based on the reviewer’s recommendation including adding an additional independent coder and consensus process.
- There was also not demographic detail or related analysis of whether any other independent variables might impact on tolerance of ambiguity.
- Demographic data was not collected as part of the pilot study and therefore is unable to be added to the analysis.
- This is a very long section relative to the others and in fact it did contain much more helpful context and reflections on the TAS and also some other insights into pharmacy and wider healthcare education. However, based on the concerns raised above, caution is needed in trying to discuss the findings given the concerns identified in the methods and results.
- We agree, and language in the last section of the discussion was added to further highlight this
Reviewer 2 Report
Comments and Suggestions for Authors
Please, see the attached document.

Author Response
Reviewer 2:
- When listing affiliations, please list the state/country.
- Revised as recommended
- Materials and methods: Why were only second professional year student pharmacists offered
to participate? How was the population for this study determined? How many students are there
in that specific year of study, in total? How many of them were approached/invited to
participate? What sampling technique was used? Sampling is not adequately described. The
authors should also address potential inclusion/exclusion criteria, if there were any.
- The IRS is only offered in this cohort of students at this level of training. Inclusion criteria included completion of the survey and completion of consent. Detail added to this section to describe the year of study and how many students were in this cohort.
- Line 69 The abbreviation IRS was only used in abstract, but not in the text itself. Please, introduce the abbreviation properly.
- IRS as an abbreviation is used throughout the manuscript. It is defined and used throughout both the abstract and the manuscript and as such no change was made
- Discussion: In line 188, the authors reference a scoping review but it is not cited here. Please, check.
- The scoping review is reference 2, referenced earlier in the paragraph, this was added per reviewer request.
- One of the main limitations of this work (in my opinion) is not mentioned among other limitations the fact that all students were from one school of Pharmacy and Pharmaceutical Science. The authors seem to ignore the fact that UT might be affected by the education process organization, educators themselves and other factors linked to specific schools.
- Caution regarding application of the results from one US institution was added to the final paragraph of the discussion.
- Overall, this work contains insufficient data to be viewed as an original article. Maybe it is better suited as short communication?
- The authors defer to the editorial team regarding whether this should be converted to a different article type.
Reviewer 3 Report
Comments and Suggestions for Authors
The topic of UT in pharmacy students using simulation is underexplored. The pilot study contributes to understanding pharmacy students’ interaction with uncertainty and proposes a practical simulation model worth exploring further. However, the novelty is limited, the sample size is small and writing need polishing.
There’s limited conceptual development beyond existing literature. For a pilot study, some form of comparison (even descriptive) to prior similar studies would help bolster its contribution.
The objectives are clearly stated (lines 57–59). However, the objective doesn’t fully reflect the depth of the qualitative analysis performed. Suggest revising to include reflection and thematic insights.
Line 120: “Self-reflection items were refined to themes” → vague. Suggest “Responses were coded and grouped into themes.”
Lack of validation of the TAS tool for this population is mentioned but not sufficiently discussed. This is a methodological limitation.
Although it’s a pilot study, No power analysis is mentioned.
The sample (n=59) is acceptable for a pilot, but response rate (48%) is low and requires justification or exploration of non-response bias.
Authors used Wilcoxon Signed-Rank test, which is appropriate for ordinal or non-normally distributed data. But no justification or test for normality is presented (e.g., Shapiro-Wilk).
No exploration of how gender, academic performance, or previous experience with uncertainty may influence UT. These covariates could reveal important insights even in pilot form. Strongly recommended to run exploratory analyses comparing TAS scores by background variables.
The coding process was led by one researcher with limited cross-checking. This risks bias in qualitative analysis.
The simulation grading rubrics could have been described or appended as supplementary material for reproducibility.
No effect sizes (e.g., Cohen’s d, r) are reported, even though the authors conclude that there was "no significant change" in TAS scores.
Table 1 formatting is problematic, hard to read. Needs formatting correction.
The thematic insights (Table 3) are meaningful and support the study’s conclusions, especially around teamwork.
The discussion is narrative-heavy and lacks visual summarization of key findings. Consider adding a diagram linking team dynamics to UT shifts.
Educational implications (curriculum design, team-based learning, use of simulations) are well-articulated.
Missed to connect UT outcomes to clinical accuracy, which is only briefly suggested at the end (line 228).
The paper doesn’t propose concrete next steps (e.g., scaling up simulation, long-term tracking).
“Insolubility” sub-scale explanation is jarring; define terms more clearly.
Adding more references would strengthen this manuscript. The existing literature on UT in other health professions is only lightly touched on. The Budner Tolerance of Ambiguity Scale is cited (1962), but more recent validation studies or psychometric analyses are not included. Add recent studies that examine how pharmacy curricula address ambiguity, decision-making, or reflective practice. Evidence linking UT with actual clinical outcomes or decision-making in pharmacy. Since “working with their team” is a major theme, it would be appropriate to cite foundational work on psychological safety
Author Response
Reviewer 3:
- There’s limited conceptual development beyond existing literature. For a pilot study, some form of comparison (even descriptive) to prior similar studies would help bolster its contribution.
- The introduction was bolstered to describe similar simulation-based interventions for medical students
- The objectives are clearly stated (lines 57–59). However, the objective doesn’t fully reflect the depth of the qualitative analysis performed. Suggest revising to include reflection and thematic insights.
- Edited per reviewer recommendation.
- Line 120: “Self-reflection items were refined to themes” → vague. Suggest “Responses were coded and grouped into themes.”
- The qualitative analysis description was rewritten and updated.
- Lack of validation of the TAS tool for this population is mentioned but not sufficiently discussed. This is a methodological limitation.
- This is mentioned in the limitations and was bolstered.
- Although it’s a pilot study, No power analysis is mentioned.
- A power analysis was not conducted as we were limited to a convenience sample of students within the course and were not able to recruit beyond that.
- The sample (n=59) is acceptable for a pilot, but response rate (48%) is low and requires justification or exploration of non-response bias.
- We have included the risk of non-response bias in our limitations section.
- Authors used Wilcoxon Signed-Rank test, which is appropriate for ordinal or non-normally distributed data. But no justification or test for normality is presented (e.g., Shapiro-Wilk).
- Given the ordinal nature of the data evaluated it is not necessary to statistically test for normality as the non-parametric Wilcoxon Signed-rank test would remain the appropriate test.
- No exploration of how gender, academic performance, or previous experience with uncertainty may influence UT. These covariates could reveal important insights even in pilot form. Strongly recommended to run exploratory analyses comparing TAS scores by background variables.
- Demographic data was not collected as part of the pilot study and therefore is unable to be added to the analysis.
- The coding process was led by one researcher with limited cross-checking. This risks bias in qualitative analysis.
- We have updated and improved our qualitative approach and reporting based on the reviewer’s recommendation including adding an additional independent coder and consensus process.
- The simulation grading rubrics could have been described or appended as supplementary material for reproducibility.
- The authors have elected not to provide these, as this was not included in the consent, and they did not address the uncertainty tolerance survey.
- No effect sizes (e.g., Cohen’s d, r) are reported, even though the authors conclude that there was "no significant change" in TAS scores.
- We have added an effect size estimate [r] to Table 2 and included more details in the Methods section.
- Table 1 formatting is problematic, hard to read. Needs formatting correction.
- Table formatting is provided in journal template.
- The thematic insights (Table 3) are meaningful and support the study’s conclusions, especially around teamwork.
- Thank you.
- The discussion is narrative-heavy and lacks visual summarization of key findings. Consider adding a diagram linking team dynamics to UT shifts.
- Educational implications (curriculum design, team-based learning, use of simulations) are well-articulated.
- Thank you.
- Missed to connect UT outcomes to clinical accuracy, which is only briefly suggested at the end (line 228).
- This was not assessed and is unable to be added to the analysis.
- The paper doesn’t propose concrete next steps (e.g., scaling up simulation, long-term tracking).
- This was added at the end of the discussion.
- “Insolubility” sub-scale explanation is jarring; define terms more clearly.
- Detail added in this section.
- Adding more references would strengthen this manuscript. The existing literature on UT in other health professions is only lightly touched on. The Budner Tolerance of Ambiguity Scale is cited (1962), but more recent validation studies or psychometric analyses are not included. Add recent studies that examine how pharmacy curricula address ambiguity, decision-making, or reflective practice. Evidence linking UT with actual clinical outcomes or decision-making in pharmacy. Since “working with their team” is a major theme, it would be appropriate to cite foundational work on psychological safety
- Additional references were added throughout the manuscript. More recent psychometric scales that could be included as the project is expanded were added in the discussion. An additional paragraph in the discussion regarding COEPA elements that may be addressed by the IRS and UT was added to strengthen the discussion in addition.
Reviewer 4 Report
Comments and Suggestions for Authors
The manuscript addresses an important and underexplored area in pharmacy education—uncertainty tolerance (UT) in student pharmacists and its potential development through simulation-based training. The research is well-aligned with ACPE 2025 Standards and responds to a growing need to better prepare future pharmacists for clinical decision-making under uncertain conditions. However, while the study is timely and the writing is clear, the manuscript has several conceptual, methodological, and reporting limitations that should be addressed to strengthen its contribution.
1.- Major Comments
1.1.- Limited Theoretical Framing of UT in Pharmacy
- The introduction references UT literature, but the discussion of its relevance to pharmacy practice lacks depth. Most cited studies are from medical education. Consider strengthening the theoretical framework by exploring how ambiguity and decision-making specifically manifest in pharmacist roles (e.g., community pharmacy, hospital settings, interprofessional teams).
1.2.- Sample Size and Generalizability
- With a sample size of 59 students (48% response rate), the results are of limited generalizability, as acknowledged. However, the lack of demographic information (e.g., gender, age, prior simulation experience) prevents deeper insights into variability in UT. Including this data (even in aggregate) would strengthen the analysis.
1.3.- Methodological Weaknesses
- Single-institution, non-randomized design and lack of a control group are major limitations. While common in pilot studies, this should be more critically discussed in the manuscript’s limitations section.
- The TAS instrument used (Budner scale) is dated and has not been validated in pharmacy students. Alternative or supplementary measures (e.g., McLain's MSTAT-II or Geller’s Tolerance for Ambiguity Scale) could provide more robust insights.
- The Wilcoxon Signed-rank test is appropriate given the data distribution, but effect sizes (e.g., r or Cohen’s d equivalents) should be reported to provide context on practical significance, especially since p-values alone can obscure meaningful findings in small samples.
1.4.- Qualitative Analysis: Depth and Rigor
- The thematic analysis lacks depth. Categories are listed with frequencies, but illustrative quotes are absent. Including 1–2 representative quotes per major theme would increase the richness and credibility of the analysis.
- The coding process involved only one primary coder and a second for consensus, but no interrater reliability metrics (e.g., Cohen's kappa) are reported. This weakens the trustworthiness of the findings. Future studies should consider triangulation or member checking.
1.5.- Interpretation of TAS Scores
- The discussion claims that higher TAS scores suggest low UT, but according to Budner (1962), lower TAS scores indicate greater tolerance for ambiguity. This interpretation may be reversed or needs clarification.
- Relatedly, the conclusion that students had “high TAS scores” and thus “low UT” contradicts standard usage of the TAS. This conceptual confusion needs immediate correction.
2.- Minor Comments
2.1.- Title: Consider clarifying that this is a "Pilot Mixed-Methods Study" to reflect the quantitative and qualitative components.
2.2.- Abstract:
- Needs better clarity on results: the TAS did not change significantly; only the insolubility subscale improved.
- P-values should be included directly in the abstract (currently only mentioned for overall TAS).
2.3.- Table Formatting:
- Tables 1 and 2 are informative but difficult to read in the current layout. Consider condensing or redesigning for clarity.
2.4.- Grammar/Stylistics:
- Minor typographical issues (e.g., inconsistent line breaks, redundant phrasing like “pre-/post-IRS”) should be corrected before publication.
2.5.- Reference List:
- Recent pharmacy-specific literature on UT is sparse but consider including broader interdisciplinary references or educational theory frameworks (e.g., cognitive load, experiential learning theory).
The study addresses an emerging and significant topic in pharmacy education. However, revisions are necessary to clarify theoretical interpretations, enhance methodological rigor, and better report and discuss qualitative findings. With these improvements, the manuscript could make a valuable contribution to simulation-based education research.
Author Response
Reviewer 4:
- Limited Theoretical Framing of UT in Pharmacy: The introduction references UT literature, but the discussion of its relevance to pharmacy practice lacks depth. Most cited studies are from medical education. Consider strengthening the theoretical framework by exploring how ambiguity and decision-making specifically manifest in pharmacist roles (e.g., community pharmacy, hospital settings, interprofessional teams).
- This was added to the introduction.
- Sample Size and Generalizability: With a sample size of 59 students (48% response rate), the results are of limited generalizability, as acknowledged. However, the lack of demographic information (e.g., gender, age, prior simulation experience) prevents deeper insights into variability in UT. Including this data (even in aggregate) would strengthen the analysis.
- Demographic data was not collected as part of the pilot study and therefore is unable to be added to the analysis.
- Single-institution, non-randomized design and lack of a control group are major limitations. While common in pilot studies, this should be more critically discussed in the manuscript’s limitations section.
- The authors believe these limitations are discussed in the limitations section of the discussion but if the editors feel this should be expanded, we can revisit.
- The TAS instrument used (Budner scale) is dated and has not been validated in pharmacy students. Alternative or supplementary measures (e.g., McLain's MSTAT-II or Geller’s Tolerance for Ambiguity Scale) could provide more robust insights.
- Limitations of the TAS instrument are discussed later in the paper. We agree that other potential scales could have been used.
- The Wilcoxon Signed-rank test is appropriate given the data distribution, but effect sizes (e.g., r or Cohen’s d equivalents) should be reported to provide context on practical significance, especially since p-values alone can obscure meaningful findings in small samples.
- We have added an effect size estimate [r] to Table 2 and included more details in the Methods section.
- Qualitative Analysis: Depth and Rigor: The thematic analysis lacks depth. Categories are listed with frequencies, but illustrative quotes are absent. Including 1–2 representative quotes per major theme would increase the richness and credibility of the analysis.
- We have updated and improved our qualitative approach and reporting based on the reviewer’s recommendation including adding an additional independent coder and consensus process.
- The coding process involved only one primary coder and a second for consensus, but no interrater reliability metrics (e.g., Cohen's kappa) are reported. This weakens the trustworthiness of the findings. Future studies should consider triangulation or member checking.
- We have updated and improved our qualitative approach and reporting based on the reviewer’s recommendation including adding an additional independent coder and consensus process.
- Interpretation of TAS Scores: The discussion claims that higher TAS scores suggest low UT, but according to Budner (1962), lower TAS scores indicate greater tolerance for ambiguity. This interpretation may be reversed or needs clarification.
- As is currently written, the interpretation is correct. Higher scores suggest lower UT.
- Relatedly, the conclusion that students had “high TAS scores” and thus “low UT” contradicts standard usage of the TAS. This conceptual confusion needs immediate correction.
- As is currently written, the interpretation is correct. Higher scores suggest lower UT.
- Title: Consider clarifying that this is a "Pilot Mixed-Methods Study" to reflect the quantitative and qualitative components.
- Added per reviewer recommendation
- Abstract:
- Needs better clarity on results: the TAS did not change significantly; only the insolubility subscale improved.
- P-values should be included directly in the abstract (currently only mentioned for overall TAS).- Edited per reviewer recommendation
- Table Formatting: Tables 1 and 2 are informative but difficult to read in the current layout. Consider condensing or redesigning for clarity.
- Tables are in journal format, expect appearance will improve with proofs.
- Grammar/Stylistics: Minor typographical issues (e.g., inconsistent line breaks, redundant phrasing like “pre-/post-IRS”) should be corrected before publication.
- Have attempted to address these where seen.
- 5.- Reference List: Recent pharmacy-specific literature on UT is sparse but consider including broader interdisciplinary references or educational theory frameworks (e.g., cognitive load, experiential learning theory).
- The discussion was bolstered, and as such, the reference list was also bolstered to address.
- The discussion was bolstered, and as such, the reference list was also bolstered to address.
Reviewer 5 Report
Comments and Suggestions for Authors
1- The introduction is well written and provides a clear justification and motivation for the study. However, it is overly brief. A more in-depth contextualization of the topic would strengthen the manuscript. Some additional elements that could be addressed include, but are not limited to: the clinical relevance of uncertainty tolerance, the value of simulation-based learning, the existing research gap (limited evidence on UT in student pharmacists and the impact of simulation on its development), and the overall study rationale.
2- The authors mention that informed consent was obtained, which implies that the activity was not part of the students’ formal curriculum—for example, they were not assessed for course credit. However, it would be prudent to state this explicitly.
3- The scoring method for the TAS items should be clarified. What numerical values were assigned to each response option on the TAS scale? According to Budner’s original version, the scale can range from 1 to 7 or from 7 to 1, depending on whether the item is positively or negatively worded. Did the authors adopt the same scoring strategy?
4- The authors highlight one TAS subscale in which a statistically significant difference was observed pre- and post-intervention, emphasizing the improvement promoted by the IRS activity. However, they do not discuss the absence of statistically significant differences in the remaining subscales or in the overall TAS score. Negative findings are still valuable results and should be addressed.
5- While the authors acknowledge the study’s limitations, I believe these significantly weaken the study and may compromise its suitability for publication. Of particular concern is the time interval between the IRS activity and the completion of the post-class TAS and self-reflection, which may have introduced a substantial recall bias.
6- The discussion section is very brief and lacks engagement with contrasting perspectives or broader literature. The excessive brevity of both the introduction and the discussion is reflected in the notably short list of references, which limits the depth and academic grounding of the manuscript.
Author Response
Reviewer 5:
- The introduction is well written and provides a clear justification and motivation for the study. However, it is overly brief. A more in-depth contextualization of the topic would strengthen the manuscript. Some additional elements that could be addressed include, but are not limited to: the clinical relevance of uncertainty tolerance, the value of simulation-based learning, the existing research gap (limited evidence on UT in student pharmacists and the impact of simulation on its development), and the overall study rationale.
- Additional detail was added to the introduction.
- The authors mention that informed consent was obtained, which implies that the activity was not part of the students’ formal curriculum—for example, they were not assessed for course credit. However, it would be prudent to state this explicitly.
- Added this information per reviewer recommendation.
- The scoring method for the TAS items should be clarified. What numerical values were assigned to each response option on the TAS scale? According to Budner’s original version, the scale can range from 1 to 7 or from 7 to 1, depending on whether the item is positively or negatively worded. Did the authors adopt the same scoring strategy?
- We have updated the methods and results section to identify the items that are reverse-socred.
- The authors highlight one TAS subscale in which a statistically significant difference was observed pre- and post-intervention, emphasizing the improvement promoted by the IRS activity. However, they do not discuss the absence of statistically significant differences in the remaining subscales or in the overall TAS score. Negative findings are still valuable results and should be addressed.
- Added this information per reviewer recommendation.
- While the authors acknowledge the study’s limitations, I believe these significantly weaken the study and may compromise its suitability for publication. Of particular concern is the time interval between the IRS activity and the completion of the post-class TAS and self-reflection, which may have introduced a substantial recall bias.
- We defer to the editors regarding this reviewer recommendation. We have edited the re-coded to come to consensus and address some of the study limitations but are unable to address the recall bias.
- The discussion section is very brief and lacks engagement with contrasting perspectives or broader literature. The excessive brevity of both the introduction and the discussion is reflected in the notably short list of references, which limits the depth and academic grounding of the manuscript.
- The discussion was bolstered, and as such, the reference list was also bolstered to address.
Round 2
Reviewer 1 Report
Comments and Suggestions for Authors
Thank you for this revision and for your response to reviewer comments and the changes made to the manuscript. There are several points of improvement and clarification which are welcome but it is noted that the authors have chosen to retain many aspects which were commented on initially. As such the paper still does not address these and this continues to give concern. There are some changes to the quantitative analysis and detail provided but this is still not optimal; the qualitative has a single supporting reference about rapid analysis which appears to come from a Powerpoint presentation and does not seem credible for publication standard. The qualitative now does have more recognisable quotes but again is presented in an overly tabulated way which appears very deductive and seems to emphasise the question domains which the authors note was simply what they decided to ask (ie no literature or theory links) and so there appears little inductive analysis. The discussion is now even longer than before which again adds to an unbalanced paper.
Sorry not to be more positive and fully respect the author's decisions on what to change and what not to change but this has resulted in a paper which is still not optimal or indeed near publication standard.
Author Response
Reviewer 1
Thank you for this revision and for your response to reviewer comments and the changes made to the manuscript. There are several points of improvement and clarification which are welcome but it is noted that the authors have chosen to retain many aspects which were commented on initially. As such the paper still does not address these and this continues to give concern. There are some changes to the quantitative analysis and detail provided but this is still not optimal; the qualitative has a single supporting reference about rapid analysis which appears to come from a and does not seem credible for publication standard. The qualitative now does have more recognisable quotes but again is presented in an overly tabulated way which appears very deductive and seems to emphasise the question domains which the authors note was simply what they decided to ask (ie no literature or theory links) and so there appears little inductive analysis. The discussion is now even longer than before which again adds to an unbalanced paper.
- We have edited the reference for the qualitative methods for rapid analysis. In regards to the other comments, we will have to defer to the editors regarding final decisions.
Reviewer 2 Report
Comments and Suggestions for Authors
I thank the authors for addressing the issues I've raised.
Author Response
Reviewer 2
- I thank the authors for addressingthe issues I've raised.
- Thank you.
Reviewer 3 Report
Comments and Suggestions for Authors
The authors responded adequately all of my comments. I have no further comments.
Author Response
Reviewer 3
- The authors responded adequately all of my comments. I have no further comments.
- Thank you.
Reviewer 4 Report
Comments and Suggestions for Authors
I appreciate the authors' thoughtful revisions and their efforts to strengthen the manuscript in response to prior feedback. This revised version demonstrates clear improvements in several key areas, including theoretical framing, statistical reporting, and qualitative rigor. However, some concerns remain either partially addressed or unresolved, particularly regarding the clarity of the TAS score interpretation, depth of the limitations section, and transparency in qualitative analysis.
1. Theoretical Framing of UT in Pharmacy Practice. The introduction now includes pharmacy-specific examples and literature, improving the relevance and contextual grounding of the study.
2. Demographic Data. Demographic data were not collected, which limits further analysis. This is clearly acknowledged in the discussion.
3. Study Design: Single Site, No Control Group, Small Sample. The manuscript mentions these limitations, but a more critical discussion of their implications (e.g., potential biases, threats to internal validity) would add rigor.
4. TAS Instrument Limitations. The discussion notes limitations of the Budner TAS but does not mention alternative instruments by name (e.g., McLain’s MSTAT-II, Geller’s scale), which would strengthen this point.
5. Effect Sizes. Effect sizes have been added to Table 2 and explained in the Methods, enhancing interpretation.
6. Qualitative Analysis. Thematic findings are now supported with illustrative quotes, and the coding process includes an additional coder and a consensus procedure.
7. Interrater Reliability Metrics. Although a second coder was involved, no interrater reliability statistic (e.g., Cohen’s kappa) is reported. This omission should at least be acknowledged.
8. Interpretation of TAS Scores. The manuscript correctly states that higher TAS scores reflect lower tolerance for ambiguity, in line with Budner (1962). However, the explanation is terse, and since the TAS measures intolerance (not tolerance) directly, this inverse relationship could confuse readers unfamiliar with the instrument. A brief conceptual clarification would strengthen consistency and reader comprehension.
9. Title Clarification. The title now includes “Pilot Mixed-Methods Study,” which reflects the study design appropriately.
10. Abstract Improvements. The abstract now includes specific p-values and clarifies the results for subscales vs. overall TAS.
11. Table Formatting .Tables remain dense but will likely be reformatted at the proof stage. No further action required at this point.
12. Stylistic and Typographical Revisions. Flow and structure have improved, though minor editorial refinements may still be beneficial in final copyediting.
13. Reference List Expansion. The revised manuscript includes a broader set of interdisciplinary references and theoretical frameworks.
Author Response
Reviewer 4
- Theoretical Framing of UT in Pharmacy Practice. The introduction now includes pharmacy-specific examples and literature, improving the relevance and contextual grounding of the study.
- Thank you.
- Thank you.
- Demographic Data. Demographic data were not collected, which limits further analysis. This is clearly acknowledged in the discussion.
- Thank you.
- Study Design: Single Site, No Control Group, Small Sample. The manuscript mentions these limitations, but a more critical discussion of their implications (e.g., potential biases, threats to internal validity) would add rigor.
- These limitations have been discussed in the first paragraph of page 10, line 319-324, and additional terms were added to bolster this section.
- TAS Instrument Limitations. The discussion notes limitations of the Budner TAS but does not mention alternative instruments by name (e.g., McLain’s MSTAT-II, Geller’s scale), which would strengthen this point.
- These are called out by name already in the discussion – page 10, line 326-328, no additional changes made at this time.
- Effect Sizes. Effect sizes have been added to Table 2 and explained in the Methods, enhancing interpretation.
- Thank you.
- Qualitative Analysis. Thematic findings are now supported with illustrative quotes, and the coding process includes an additional coder and a consensus procedure.
- Thank you.
- Interrater Reliability Metrics. Although a second coder was involved, no interrater reliability statistic (e.g., Cohen’s kappa) is reported. This omission should at least be acknowledged.
- This was added to the limitations section: “While we used multiple coders, no formal assessment of interrater reliability was calculated.”
- Interpretation of TAS Scores. The manuscript correctly states that higher TAS scores reflect lower tolerance for ambiguity, in line with Budner (1962). However, the explanation is terse, and since the TAS measures intolerance (not tolerance) directly, this inverse relationship could confuse readers unfamiliar with the instrument. A brief conceptual clarification would strengthen consistency and reader comprehension.
- The interpretation of TAS scores is discussed on page 8, lines 230-232. In addition, during the last revision round, a description of the scoring and how it is interpreted was added to the methods and in the footnotes of Table 1.
- Title Clarification. The title now includes “Pilot Mixed-Methods Study,” which reflects the study design appropriately.
- Thank you.
- Abstract Improvements. The abstract now includes specific p-values and clarifies the results for subscales vs. overall TAS.
- Thank you.
- Table Formatting .Tables remain dense but will likely be reformatted at the proof stage. No further action required at this point.
- Thank you.
- Stylistic and Typographical Revisions. Flow and structure have improved, though minor editorial refinements may still be beneficial in final copyediting.
- Thank you.
- Reference List Expansion. The revised manuscript includes a broader set of interdisciplinary references and theoretical frameworks.
- Thank you.
Reviewer 5 Report
Comments and Suggestions for Authors
I appreciate the revisions made to the manuscript. The updated version demonstrates clearer contextualization, improved methodological detail, and a more robust discussion, all of which contribute to strengthening the overall quality of the work. The inclusion of both statistically significant and non-significant findings is also a valuable addition.
However, I maintain some reservations regarding the issue of recall bias, stemming from the time gap between the IRS activity and the administration of the post-intervention TAS and self-reflection measures. This limitation may undermine the reliability of the data and, consequently, the validity of the study’s conclusions. It represents a methodological flaw that could have been avoided with more rigorous planning of data collection. While I acknowledge the authors’ efforts to address several study limitations, I consider the recall bias a critical concern that should be carefully weighed in the editorial decision.
Author Response
Reviewer 5
- I appreciate the revisions made to the manuscript. The updated version demonstrates clearer contextualization, improved methodological detail, and a more robust discussion, all of which contribute to strengthening the overall quality of the work. The inclusion of both statistically significant and non-significant findings is also a valuable addition.
- Thank you.
- However, I maintain some reservations regarding the issue of recall bias, stemming from the time gap between the IRS activity and the administration of the post-intervention TAS and self-reflection measures. This limitation may undermine the reliability of the data and, consequently, the validity of the study’s conclusions. It represents a methodological flaw that could have been avoided with more rigorous planning of data collection. While I acknowledge the authors’ efforts to address several study limitations, I consider the recall bias a critical concern that should be carefully weighed in the editorial decision.
- Due to the nature of the course, half of the students completed the interprofessional rounding simulation in one week and the other half completed it in the second week. As such, the study was kept open for two weeks to capture both cohorts of students. As this was a pilot study, this limitation is acknowledged and future directions noted; page 10, line 331-334. We will have to defer to editors regarding the final decision.